# Multi-Label Generalized Zero Shot Chest Xray Classification Using Feature Disentanglement and Multi-Modal Dictionaries

## Abstract

Generalized zero shot learning (GZSL) aims to correctly predict seen and unseen classes, and most GZSL methods focus on the single label case. However, medical images can have multiple labels as in the case of chest x-rays. We propose a novel multi-modal multi-label GZSL approach that leverages feature disentanglement and multi-modal dictionaries to synthesize features of unseen classes. Feature disentanglement extracts class specific features, which are used with text embeddings to learn a multi-modal dictionary. A subsequent clustering step identifies class centroids, all of which contribute to better multi-label feature synthesis. Compared to existing methods, our approach does not require class attribute vectors, which are an essential part of GZSL methods for natural images but are not available for medical images. Our approach outperforms state of the art GZSL methods for chest x-rays. We also analyse the performance of different loss terms in ablation studies.

## 1 Introduction

Fully supervised deep learning methods provide state-of-the-art (SOTA) performance for a variety of medical image analysis tasks Gulshan et al. (2016); Irvin et al. (2017) due to access to all classes during training. In real radiological workflows unseen disease types are commnly encountered, e.g., new strains of COVID-19 or tumour types in histopathological data. Conventional fully supervised approaches misclassify the new disease subtypes into one of previously seen classes leading to erroneous diagnosis and lengthy system re-certification loops for clinically deployed AI systems.

Zero-Shot Learning (ZSL) aims to learn plausible representations of unseen classes from seen class features. In a more generalized setting of Generalized Zero-Shot Learning (GZSL), we expect to encounter both seen and unseen classes during the test phase. Previous works on GZSL in medical images have mostly focused on the single label scenario Mahapatra et al. (2022); Paul et al. (2021). However, chest X-ray (CXR) datasets have multiple labels assigned to the images and single-label methods do not work well in this setting. Hayat et al. (2021) proposed a multi-label GZSL method to predict multiple seen and unseen diseases in CXR images by mapping visual and semantic modalities to a latent feature and learning a visual representation guided by the input's corresponding semantics extracted from a medical text corpus. They obtain sub-optimal results on the external NIH chest xray dataset Wang et al. (2017) in terms of AUROC values of seen (0.79) and unseen (0.66) classes, possibly due to the sub-optimal use of multi-label text and imaging data. We propose a multi-label GZSL approach that uses multi-modal dictionaries encoding text and imaging information to encode the semantic relationship between multiple disease labels. This enables us to learn a highly accurate feature representation which plays an important role in synthetic feature generation.

In contrast with medical imaging datasets for GZSL, datasets for GZSL in natural images Feng et al. (2022); Lee et al. (2018); Su et al. (2022); Kong et al. (2022) have the advantage of providing attribute vectors for all classes to enable a model to correlate between attribute vectors and corresponding feature representations of the seen classes. Defining unambiguous attribute vectors for medical images requires deep clinical expertise and extensive invested time to annotate radiological images. This complexity is exacerbated for the multi-label scenario, where many disease conditions have similar appearances and textures.

To achieve this non-trivial task we introduce the following contributions for multi-label GZSL: **1**) We propose a novel feature disentanglement method where a given image is decomposed into class-specific and class-agnostic features. This step is necessary for multi-label problems to enable accurate combination of class-specific features. **2**) We apply graph aggregation on class-specific features to learn an image feature based multi-label dictionary based on interactions between different labels at a global scale. This leads to more discriminative feature learning and contributes to better multi-label feature synthesis. **3**) We learn the semantic relationships between text embeddings of different disease class labels and use this knowledge to guide the generation of realistic feature vectors that preserve the semantic relationship among multiple disease labels.

## 2 Prior Work

**Feature Disentanglement In Medical Image Analysis:** Liu et al. (2022) provide a comprehensive overview of feature disentanglement techniques in medical image analysis. Wang et al. (2023) propose a feature disentanglement based unsupervised domain adaptation (UDA) method for image segmentation and apply it to retinal vessel segmentation. Chartsias et al. (2019) propose Spatial Decomposition Network (SDNet) to decompose 2D medical images into spatial anatomical factors and non-spatial modality factors. They use it for different cross modal segmentation tasks. Ouyang et al. (2021) use margin loss, conditional convolution and a fusion function, with applications to three multi-modal neuroimaging datasets for brain tumor segmentation.

**Generalized Zero-Shot Learning:** In GZSL, the purpose is to recognize images from known and unknown domains. Prior work on natural images show promising results by training GANs in the known domain and generating unseen visual features from semantic labels Felix et al. (2018); Verma et al. (2018); Xian et al. (2019). Keshari et al. (2020) use over-complete distributions to generate features of unseen classes, while Min et al. (2020) used domain-aware visual bias elimination for synthetic feature generation. Feng et al. (2022) propose a non-generative model for synthesizing edge-pseudo and center-pseudo samples to introduce greater diversity. The work by Kong et al. (2022) promotes intra-class compactness with inter-class separability on both seen and unseen classes in the embedding space and visual feature space. Su et al. (2022) leverage visual and semantic modalities to distinguish seen and unseen categories by deploying two variational autoencoders to generate latent representations for visual and semantic modalities in a shared latent space.

**Multi-Label Zero-Shot Learning:** Lee et al. (2018) propose a novel deep learning architecture for multi-label zero-shot learning (ML-ZSL), that predicts multiple unseen class labels for each input instance using an information propagation mechanism from the semantic label space. Zhang et al. (2016) consider the separability of relevant and irrelevant labels, proposing a model that learns principal directions for images in the embedding space. Gaure et al. (2017) leverage co-occurrence statistics of seen and unseen labels and learns a graphical model for ZSL.

**GZSL In Medical Images:** This is a less explored topic primarily because conventional methods from the natural image domain cannot be directly applied due to lack of class attribute vectors for medical images. Some initial works explored registration Kori & Krishnamurthi (2019) and artifact reduction Chen et al. (2020). Paul et al. (2021) proposed a GZSL method for chest X-ray diagnosis by learning the relationship between multiple semantic spaces (from X-ray, CT images, and reports). However, not all datasets have multiple image modalities and text reports. Mahapatra et al. (2022) proposed a class-attribute-free method for GZSL on different medical images by using saliency maps and self-supervised learning. Hayat et al. (2021) learn an image's visual representation guided by the input's corresponding semantics extracted from BioBERTLee et al. (2019), a BERT Devlin et al. (2018)-based language model. The primary challenge of multi-label GZSL is to synthesize features that capture the characteristics of multiple classes. Different from other works, our method, combining feature disentanglement with image and text features, works with images from a single modality and shows state-of-the-art performance on multiple public CXR datasets.

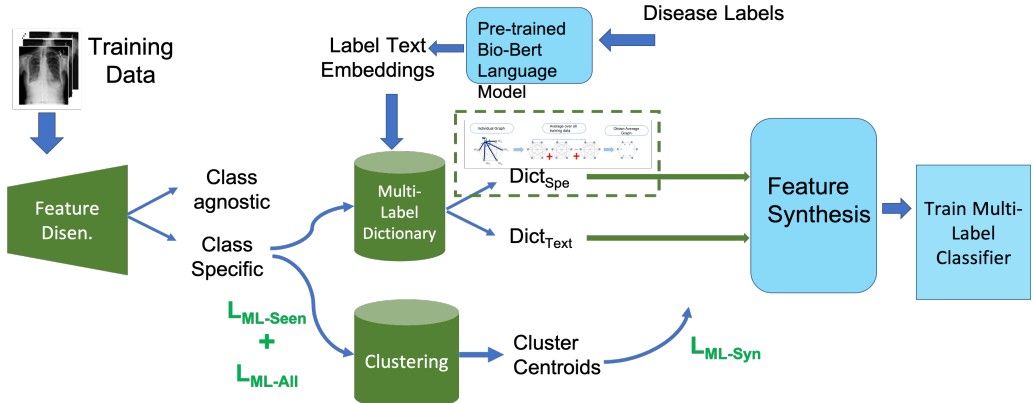

Figure 1: Workflow of the proposed method. Training data goes through a feature disentanglement stage, followed by multi-modal and multi-label dictionary learning and clustering, feature synthesis and training of a multi-label classifier. Our novel contributions and loss functions are highlighted as green blocks and letters. $Dic_{Spe}$ is the dictionary created from class specific features of seen classes and $Dic_{Text}$ is the dictionary obtained from label texts.

## 3 METHOD

**Method Overview:** Figure 1 depicts our proposed workflow with the following stages: 1) Image feature disentanglement to get class-specific and class-agnostic features; 2) Create two multi-label dictionaries using class-specific features and text embeddings of disease labels. Class-specific features learn a global relationship between label features, whereas the text embeddings for different labels are obtained from BioBert Lee et al. (2019). We use this dictionary to guide the clustering and feature synthesis steps; 3) Clustering of seen and unseen class samples to obtain class centroids that function as class representative vectors; 4) Feature synthesis to generate multi-label features of different label combinations. The centroid vectors are used as reference vectors for feature synthesis. The synthesized vectors are compared with the centroids to determine whether they belong to the desired classes; 5) Train a multi-label classifier using synthesized and real features of unseen and seen classes.

### 3.1 FEATURE DISENTANGLEMENT

Feature disentanglement for domain adaptation separates the features into domain-specific and domain-invariant components Park et al. (2020). In case of GZSL the data is from the same domain with different labels. Hence we propose to decompose the feature space of seen class samples into 'class-specific' and 'class-agnostic' features. The class-specific features of each class will encode information specific to the particular class, and the class-specific features of different classes will be dissimilar. On the other hand, the class-agnostic features (e.g., characterization of bone in X-ray scans) of each class will be highly similar to each other. In this setup, we aim to yield class-specific and class-agnostic features to be mutually complementary and hence have minimal overlap in semantic content. Feature disentanglement helps to obtain features specific to each class which in turn allows for more accurate synthesis of multi-label features by incorporating characteristics of the desired classes.

Figure 2 shows the architecture of our feature disentanglement network (FDN). The FDN consists of $L$ encoder-decoder architectures corresponding to the $L$ classes in the training data. The encoders and decoders (generators) are denoted, respectively, as $E_l(\cdot)$ and $G_l(\cdot)$. Similar to a classic autoencoder, the encoder, $E_l$, produces a latent code $z_i$ for image $x_i \sim p$. Each decoder, $G_l$, reconstructs the original image from $z_i$. Furthermore, we divide the latent code, $z_i$, into two components: a class-specific component, $z_i^{spec_l}$ for class $l$, and a class-agnostic component, $z_i^{agn_l}$. Both components are vectors, and they are combined and fed to the decoder, which reconstructs the original input. The disentanglement network is trained using the following loss function:

$$\mathcal{L}_{Disent} = \mathcal{L}_{Rec} + \lambda_1 \mathcal{L}_{spec} + \lambda_2 \mathcal{L}_{agn} + \lambda_3 \mathcal{L}_{agn-spec} \qquad (1)$$

**Reconstruction Loss**: $\mathcal{L}_{Rec}$, is the commonly used image reconstruction loss and is defined as:

$$\mathcal{L}_{Rec} = \sum_{l=1}^{L} \mathbb{E}_{x_i \sim p_l} \left[ \left\| x_i^l - G_l(E_l(x_i^l)) \right\| \right] \tag{2}$$

The above term is a sum of the reconstruction losses from the class specific autoencoders. We train different autoencoders for each class in order to obtain class specific features.

**Class Specific Loss**: For given class $l$ the class specific component $z_i^{spec_l}$ will have high similarity according to some metric (e.g. cosine similarity) with samples from the same class. Since this feature is class specific it will have low similarity with the $z_i^{spec_k}$ of other classes $k$ ($k \neq l$). These two conditions are incorporated using the following terms

$$\mathcal{L}_{spec} = \sum_{i,j} \sum_{l} \left( 1 - \langle z_i^{spec_l}, z_j^{spec_l} \rangle \right) + \sum_{k} \langle z_i^{spec_l}, z_j^{spec_k} \rangle \tag{3}$$

where $\langle . \rangle$ denotes cosine similarity. The first term encourages high similarity for class specific features of samples having the same training labels. The second term encourages different classes, $l$ and $k$ to have highly dissimilar class specific features. The sum is calculated for all classes indexed by $\sum_l$ and over all samples indexed by $i, j$.

**Class Agnostic Loss**: The class agnostic features of different classes have, by definition, similar semantic content and hence they will have high cosine similarity. $L_{agn}$ is defined as

$$\mathcal{L}_{agn} = \sum_{i,j} \sum_{l} \sum_{k} \left( 1 - \langle z_i^{agn_l}, z_j^{agn_k} \rangle \right) \tag{4}$$

The above formulation ensures that the loss is minimized. Finally, we want the class specific and class agnostic features of same-class samples to be mutually complementary and have minimal overlap in semantic content. This implies that their cosine similarity values should be minimal. Hence the final loss term is defined as

$$\mathcal{L}_{agn-spec} = \sum_{l} \langle z_i^{agn_l}, z_j^{spec_l} \rangle \tag{5}$$

Since the above loss terms are minimized it helps us achieve our stated objectives.

Figure 3 (a) shows the t-sne plots of image features (taken from the fully connected layer of a DenseNet-121 trained for image classification) while Figure 3 (b) shows the plot using the class specific features. We observe that plots of the original features shows different image class clusters that overlap and that makes it challenging to have good classification. On the other hand the clusters obtained using the class specific features are well separated and there is less overlap between different clusters. Figure 3 (c) shows the output of using class agnostic features where a significant overlap is observed between classes. This clearly demonstrates the efficacy of our feature disentanglement method, i.e., the class specific and class agnostic features fulfil their desired objectives. The features are taken from images belonging to 5 classes from the NIH dataset. We chose 5 classes to clearly demonstrate the output since more classes clutter the figure.

## 3.2 Dictionary Of Text Embeddings

We generate embeddings of image class labels using BioBERT Lee et al. (2020), a BERT Devlin et al. (2018)-like pre-trained model. BioBERT Lee et al. (2020) is pre-trained on biomedical literature, more specifically the model available from Huggingface[1], which is a base and cased model. We consider a pooled set that produces a single 768 dimension vector for a label. We then calculate the cosine similarity between each of the labels and is represented as a matrix, which we refer as $Dict_{Text}$ - dictionary for text embeddings (Figure 2 (b)).

## 3.3 Multi-Modal Multi-Label Dictionary

The dictionary is constructed from two sources: 1) class specific features of seen class samples; 2) text embeddings of label vectors for all classes ($Dict_{text}$ described in Section 3.2). We learn

---

[1] https://huggingface.co/dmis-lab/biobert-v1.1

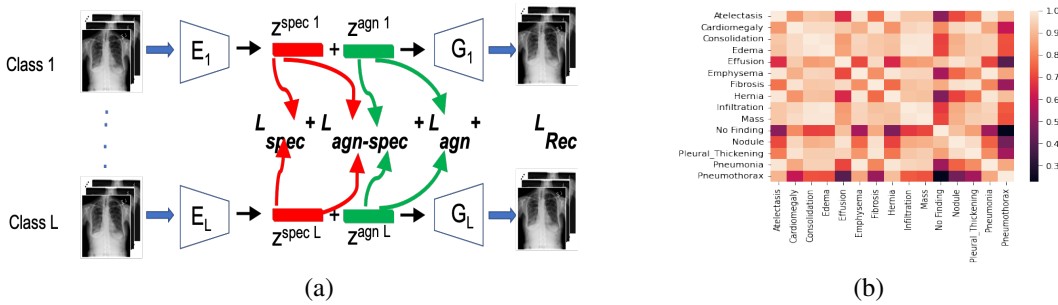

(a)                                                                (b)

Figure 2: (a) Architecture of class specific feature disentanglement network. Given training images from different classes of the same domain we disentangle features into class specific and class agnostic using autoencoders. The different feature components are used to define the different loss terms. (b) Cosine similarity of the labels' BioBERT embeddings

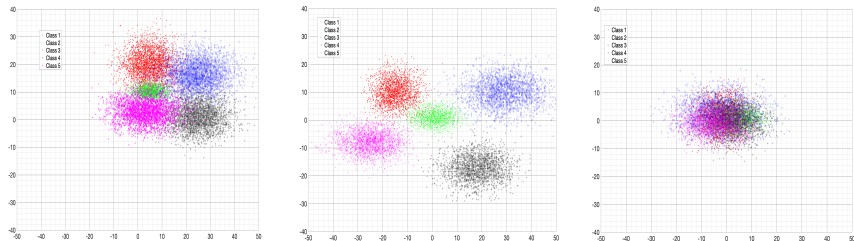

Figure 3: T-sne results comparison between original image features and feature disentanglement output. (a) Original image features; (b) Class specific features; (c) Class agnostic features.

image feature based dictionary only of seen classes. Since we do not know the actual image samples of unseen classes it is difficult to identify features corresponding to them and construct the corresponding dictionary. We construct a graph from the **seen** class samples in the following manner: **1:** Represent each image sample as a separate graph. **2:** Within a graph each Seen class label (representing a disease or condition) is represented by a node which is represented using the class specific features. Since we train multiple feature disentanglement network corresponding to number of classes we obtain, for each image, a class specific feature for all disease labels (nodes). **3:** Edge weights in the graph represent the similarity between corresponding nodes using cosine similarity of class specific features. Assuming $K$ nodes in each graph (i.e., $K$ seen classes), each node has $K - 1$ edge weights to all other nodes. The edge weight $w_{ij}$ between nodes $i, j$ is $w_{ij} = cosine\_similarity(z_I^{spec_l}, z_I^{spec_k}) = \langle z_I^{spec_l}, z_I^{spec_k} \rangle$; where $z_I^{spec_l}$ and $z_I^{spec_k}$ are the class specific features, respectively, of classes $l$ and $k$ for sample images $I$. Each graph has $\frac{K(K-1)}{2}$ edges.

**Informativeness Dictionary:** We average the inter-node weights across all graphs, to get an 'average' graph. Each inter-node link value quantifies the average cosine similarity across all training samples from the seen class. An example inter-node similarity is depicted in Table 1, and we refer to this matrix as $Dict_{Spe}$ the multi-label dictionary from class specific features. Any synthetically generated sample will preserve this relationship between Seen labels by using appropriate losses.

## 3.4 SSL Based Clustering

Having created multi-label dictionaries from two modalities our next step is to synthesize multi-label features that will play an important role in training the classifier to recognize seen and unseen classes. Before that we determine the centroids of different class specific features which function as reference vectors (or *class anchor vectors* for individual classes Li et al. (2019)) to determine whether synthesized features have characteristics of the desired classes. We use class specific features, $z^{spec_l}$, and apply self supervised learning (SSL) based online clustering approach SwAV (**Sw**apping **A**ssignments between multiple **V**iews) Caron et al. (2020) to determine the seen class

|  | Atelectasis | Cardiomegaly | Consolidation | Edema | Effusion |
|---|---|---|---|---|---|
| Atelectasis | 1 | 0.80 | 0.89 | 0.90 | 0.68 |
| Cardiomegaly | 0.80 | 1 | 0.91 | 0.91 | 0.89 |
| Consolidation | 0.89 | 0.91 | 1 | 0.94 | 0.80 |
| Edema | 0.90 | 0.91 | 0.94 | 1 | 0.83 |
| Effusion | 0.68 | 0.89 | 0.80 | 0.83 | 1 |

Table 1: Example of the multi-label similarity dictionary from saliency maps for seen classes only. This is an example dictionary for K=5 seen classes.

centroids. Our experimental results in Figure 3 show clustering using class specific features results in better separability of the clusters than using image features from pre-trained feature extractors.

Let the number of seen and unseen classes be, respectively, $n_S$ and $n_U$. We first cluster seen class features into $n_S$ clusters and obtain their centroids as $C_S = c_1, \cdots, c_{n_S}$. We enforce the constraint that the semantic relationship between the Seen class centroids be close to that obtained from $Dict_{Spe}$. This is achieved by constructing a matrix of inter-label similarities using the cosine distance between the cluster centroids at each iteration, denoted as $Cent_{Seen}(i,j)$. We then calculate an element wise difference between $Dict_{Spe}$ and $Cent_{Seen}(i,j)$:

$$L_{ML-Seen} = \frac{1}{n_S^2} \sum_i \sum_j Dict_{Spe}(i,j) - Cent_{Seen}(i,j). \qquad (6)$$

Since the matrix of cosine similarities is a square matrix having $n_S$ rows and columns it is divided by a factor of $n_S^2$ to get a normalized distance measure. $L_{ML-Seen}$ is the loss for multi-label **seen** classes. In the next pass, we compute the clusters $C_U = c_{n_S+1}, \cdots, c_{n_S+n_U}$ of the $n_U$ unseen classes using the following additional constraints:

1. The centroids in $C_S$ are kept fixed. Since the centroids $C_S$ have been computed from labeled samples we assume that the computed centroids are reliable and are not changed in the second stage.

2. We add a constraint that the semantic relationship between the seen and unseen class centroids should follow the dictionary $Dict_{Text}$ created using the text embedding vectors as described in Section 3.2. This condition is implemented using:

$$\mathcal{L}_{ML-All} = \frac{1}{N^2} \sum_i \sum_j Dict_{Text}(i,j) - Cent_{All}(i,j) \qquad (7)$$

where $Cent_{All}$ refers to the changing matrix of cluster centroid similarities for all seen and unseen classes. $N = n_S + n_U$ is the total number of seen and unseen classes.

Given image features $x_t$ and $x_s$ from two different transformations of the same image, we compute their cluster assignments $q_t$ and $q_s$ by assessing the distance of the features to a set of $K$ cluster centers $c_1, \cdots, c_K$. A "swapped" prediction problem Caron et al. (2020) is solved using :

$$\mathcal{L}(x_t, x_s) = \ell(x_t, q_s) + \ell(x_s, q_t), \qquad (8)$$

where $\ell(x, q)$ measures the fit between features $x$ and assignment $q$. Thus we compare features $x_t$ and $x_s$ using their intermediate cluster assignments $q_t$ and $q_s$. If the two $x$'s capture same information, we can predict the cluster assignment from the other feature. The final loss term for **clustering** all class samples is

$$\mathcal{L}_{Clust} = \mathcal{L}(x_s, x_t) + \lambda_4 \mathcal{L}_{ML-Seen} + \lambda_5 \mathcal{L}_{ML-All} \qquad (9)$$

We obtain a set of cluster centroids for seen and unseen classes to guide the feature generation step.

### 3.5 FEATURE GENERATION NETWORK

We synthesize the class specific features of unseen and seen classes following Xian et al. (2018). Given training images of seen classes, and unlabeled images of the unseen classes we learn a generator $G : \mathcal{E}, \mathcal{Z} \longrightarrow \mathcal{X}$, which takes a class label vector $e^y \in \mathcal{E}$ and a Gaussian noise vector $z \in \mathcal{Z}$

as inputs, and generates a feature vector $\tilde{x} \in \mathcal{X}$. The discriminator $D : \mathcal{X}, \mathcal{E} \rightarrow [0, 1]$ takes a real feature $x$ or synthetic feature $\tilde{x}$ and corresponding class label vector $e^y$ as input and determines whether the feature vector matches the class label vector. $G$ aims to fool $D$ by producing features highly correlated with $e^y$ using a Wasserstein adversarial lossArjovsky et al. (2017):

$$\mathcal{L}_{WGAN} = \min_G \max_D \mathbb{E}[D(x, e^y)] - \mathbb{E}[D(\tilde{x}, e^y)]$$
$$- \lambda \mathbb{E}[(\|\nabla_{\tilde{x}} D(\tilde{x}, e^y)\|_2 - 1)^2], \quad (10)$$

where the third term is a gradient penalty term, and $\tilde{x} = \alpha x + (1 - \alpha)\tilde{x}$. $\alpha \sim U(0, 1)$ is sampled from a uniform distribution.

$D$ is a classifier that determines whether the generated feature vector $\tilde{x}$ belongs to one of the seen classes. As the anchor vectors (i.e., the cluster centers) are fixed, we calculate the cosine similarity between the generated vector $\tilde{x}$ and the anchor vector corresponding to the desired classes. Since we are synthesizing multi-label features it is expected that the cosine similarities of the synthetic vector will be high with respect to the centroids of the desired classes. We integrate these conditions in the following formulation:

$$\mathcal{L}_{ML-Syn} = \sum_{l_y} (1 - \langle \tilde{x}, c_y \rangle) \quad (11)$$

$\mathcal{L}_{ML-Syn}$ is the multi-label synthetic loss. If $\tilde{x}$ truly represents the set of desired classes $y$ then the cosine similarity between $\tilde{x}$ and the corresponding anchor vectors $c_y$ is high and the loss is low.

### 3.6 TRAINING, INFERENCE AND IMPLEMENTATION

The final loss function for **feature generation** is:

$$\mathcal{L} = \mathcal{L}_{WGAN} + \lambda_6 \mathcal{L}_{ML-Syn} \quad (12)$$

where $\lambda_6$ is a weight balancing the contribution of the different terms. Once training is complete we specify the label of desired classes and input a noise vector to $G$ which synthesizes a new feature vector. We combine the synthesized target features of the unseen classes $\tilde{x}^u$ and real and synthetic features of seen class $x^s, \tilde{x}^s$ to construct the training set. Then we train a multi-label sigmoid classifier by minimizing the negative log likelihood loss:

$$\min_\theta -\frac{1}{|\mathcal{X}|} \sum_{(x,y) \in (\mathcal{X}, \mathcal{Y})} \log P(y|x, \theta), \quad (13)$$

where $P(y|x, \theta) = \frac{\exp(\theta_y^T x)}{1 + \exp(\theta_y^T x)}$ is the classification probability and $\theta$ denotes classifier parameters.

**Inference:** Given initial seen and unseen class samples, the clustering stages yields class centroids. The subsequent feature synthesis module generates samples of different classes for classifier training, and applying to test features.

**Implementation Details:** We compare the results of our method for medical images with existing GZSL methods. For methods developed for natural images we replace the class label vector $e^y$ with the corresponding class attribute vectors. For feature extraction we use our feature disentanglement approach to obtain class specific features. The generator (G) and discriminator (D) are all multilayer perceptrons. $G$ has two hidden layers of 2000 and 1000 units respectively while the discriminator D is implemented with one hidden layer of 1000 hidden units. We chose Adam Kingma & Ba (2014) as our optimizer, and the momentum was set to $(0.9, 0.999)$. The values of loss term weights are $\lambda_{CL} = 0.6, \lambda_3 = 0.9$. Training the Swav Clustering algorithm takes 12 hours and the feature synthesis network for 50 epochs takes 17 hours, all on a single NVIDIA V100 GPU (32 GB RAM). PyTorch was used for all implementations.

### 3.7 EVALUATION PROTOCOL

The seen class $S$ can have samples from 2 or more disease classes, and the unseen class $U$ contains samples from the remaining classes. We use all possible combinations of labels in $S$ and $U$. Following standard practice for GZSL, average class accuracies are calculated for two settings: 1) **S**: training is performed on synthesized samples of $S + U$ classes and test on the seen test set $S_{Te}$. 2)

**U**: training is performed on synthesized samples of $S + U$ classes and test on unseen test set $U_{Te}$. We also report the harmonic mean defined as,

$$H = \frac{2 \times Acc_U \times Acc_S}{Acc_U + Acc_S} \tag{14}$$

where $Acc_S$ and $Acc_U$ denote the accuracy of images from seen (setting $S$) and unseen (setting $U$) classes respectively:

# 4 EXPERIMENTAL RESULTS

## 4.1 DATASET DESCRIPTION

We demonstrate our method's effectiveness on the following chest xray datasets for multi-label classification tasks: **1. NIH Chest X-ray** Dataset: For lung disease classification we adopted the NIH Chest X-ray14 dataset Wang et al. (2017) having $112,120$ expert-annotated frontal-view X-rays from $30,805$ unique patients and has 14 disease labels. Hyperparameter values are $\lambda_1 = 1.1, \lambda_2 = 0.7, \lambda_3 = 0.9, \lambda_4 = 1, \lambda_5 = 1.1, \lambda_6 = 1.1$. **2. CheXpert** Dataset: We used the CheXpert dataset Irvin et al. (2017) consisting of $224,316$ chest radiographs of $65,240$ patients labeled for the presence of 14 common chest conditions. Hyperparameter values are $\lambda_1 = 1.2, \lambda_2 = 0.8, \lambda_3 = 1.1, \lambda_4 = 0.9, \lambda_5 = 1.2, \lambda_6 = 0.9$. For both datasets original images were resized to $224 \times 224$, and the reported results are an average of 25 runs across different combinations. A $70/10/20$ split at patient level was done to get training, validation and test sets for both datasets.

**Comparison Methods:** We compare our method's performance with multiple GZSL methods - single label and multi-label techniques - employing different feature generation approaches such as CVAE or GANs. Our method is denoted as ML-GZSL (**M**ulti **L**abel **GZSL**).

| Method | NIH X-ray | | | CheXpert | | | | | NIH X-ray | | | CheXpert | | |
|---|---|---|---|---|---|---|---|---|---|---|---|---|---|---|
| | S | U | H | S | U | H | | | S | U | H | S | U | H |
| **Single Label GZSL Methods** | | | | | | | | **ML-GZSL** | **Proposed Method And Benchmark** | | | | | |
| SDGN Wu et al. (2020) | 84.4 | 81.1 | 82.7 | 89.8 | 88.3 | 89.0 | | **ML-GZSL** | **86.2** | **84.8** | **85.5** | **90.8** | **90.2** | **90.5** |
| Feng Feng et al. (2022) | 84.7 | 81.4 | 83.0 | 90.2 | 88.6 | 89.4 | | FSL(Multi Label) | 86.0 | 85.1 | 85.5 | 90.8 | 90.5 | 90.6 |
| Kong Kong et al. (2022) | 84.8 | 81.2 | 82.9 | 90.0 | 88.7 | 89.3 | | | **Feature Disentanglement Effects** | | | | | |
| Su Su et al. (2022) | 84.5 | 81.4 | 82.9 | 90.3 | 88.6 | 89.4 | | $w\mathcal{L}_{agn-spec}$ | 83.8 | 81.9 | 82.8 | 88.6 | 86.3 | 87.4 |
| **Different Clustering Methods** | | | | | | | | pre-train | 83.4 | 82.0 | 82.7 | 88.2 | 85.3 | 86.7 |
| Deep-Cluster Caron et al. (2018) | 83.9 | 80.7 | 82.2 | 88.9 | 87.4 | 88.1 | | $w\mathcal{L}_{agn}$ | 84.5 | 82.1 | 83.3 | 89.1 | 86.9 | 88.0 |
| K-Means | 83.4 | 80.7 | 82.0 | 88.2 | 87.2 | 87.7 | | $w\mathcal{L}_{spec}$ | 83.0 | 81.2 | 82.1 | 87.8 | 85.2 | 86.5 |
| **Multi Label GZSL Methods** | | | | | | | | | **Effect of Dictionary/Clustering** | | | | | |
| Hayat Hayat et al. (2021) | 79.1 | 69.2 | 73.8 | 81.2 | 79.8 | 80.5 | | $w\mathcal{L}_{Dict_{Text}}$ | 83.2 | 81.0 | 82.1 | 87.6 | 85.1 | 86.3 |
| Lee Lee et al. (2018) | 85.1 | 81.3 | 83.1 | 87.4 | 85.7 | 86.5 | | $w\mathcal{L}_{Dict_{Spc}}$ | 82.6 | 80.7 | 81.6 | 87.0 | 84.5 | 85.7 |
| Huynh Huynh & Elhamifar (2020) | 84.7 | 80.8 | 82.7 | 86.9 | 85.1 | 86.0 | | $w\mathcal{L}_{ML-Syn}$ | 81.3 | 79.9 | 80.6 | 85.8 | 82.8 | 84.2 |

Table 2: **GZSL Results For CXRs in Multi-Label setting:** Average per-class classification accuracy (%) and harmonic mean accuracy (H) for GZSL. FSL denotes the benchmark fully supervised learning approach. Results demonstrate the superior performance of our proposed method.

## 4.2 GENERALIZED ZERO SHOT LEARNING RESULTS

Classification results for CXRs in Table 2 show our proposed method significantly outperforms all competing GZSL methods. This significant difference in performance can be explained by the fact that the complex architectures that worked for natural images will not be equally effective for medical images which have less information. Our proposed ML-GZSL method does as good as the multi-label fully supervised learning (FSL) benchmark (using a DenseNet-121 classifier). Class specific features play an important role here since they focus on the features relevant to specific classes and provide more discriminatory information than an FSL approach. We also show results when using different clustering methods such as Deep-Cluster and k-means instead of SwAV, with our feature generation method. The results are inferior to our proposed method thus demonstrating the fact that SwAV gives better representations of the cluster centroids.

## 4.3 ABLATION STUDIES FOR MEDICAL IMAGES

Table 2 also shows results for ablation studies, which are grouped under two categories: 1) Feature disentanglement and 2) Clustering using multi-label dictionaries. For the ablation methods related to

feature disentanglement we exclude each of the three loss terms - $\mathcal{L}_{agn}, \mathcal{L}_{spec}$ and $\mathcal{L}_{agn-spec}$ - and report the results as ML-GZSL$_{w\mathcal{L}_{agn}}$, ML-GZSL$_{w\mathcal{L}_{spec}}$, and ML-GZSL$_{w\mathcal{L}_{agn-spec}}$. We also compare with the results of using image features obtained from a CNN based feature extractor (ResNet50 trained on Imagenet), which we denote as 'pre-train'. We observe that the class specific features has the greatest influence on the results and excluding it, ML-GZSL$_{w\mathcal{L}_{spec}}$ results in maximum degradation of performance compared to ML-GZSL. ML-GZSL$_{w\mathcal{L}_{agn-spec}}$ shows the next worst performance, while ML-GZSL$_{w\mathcal{L}_{agn}}$ shows the least difference among the three methods. These results highlight the importance of the class specific features and at the same time illustrate that the class agnostic features have a relatively smaller influence on the method's performance. This is desirable since our objective for feature disentanglement was to get complementary features.

The second category of ablation experiments are related to learning the multi-modal multi-label dictionary, clustering and feature synthesis. The primary goal of dictionary learning is to influence clustering and feature synthesis. We conduct two set of experiments where we exclude Dict$_{Spe}$ and Dict$_{Text}$. The ablation study results in Table 2 show the two dictionaries have similar influences on the outcome, with Dict$_{Text}$ exerting a greater influence due to its ability to encode more information from all classes. Excluding $\mathcal{L}_{ML-Syn}$ uses only the Wasserstein loss for feature synthesis without including the class centroids. This results in a significant performance degradation since there is no mechanism to check the realism of synthetic features. This leads to a severe reduction in performance as the classifier is trained with lots of spurious samples which affect the final performance.

## 4.4 HYPERPARAMETER SELECTION

Figure 4 shows the harmonic mean values for the NIH Chest X-ray dataset for different values of hyperparameters $\lambda_1, \lambda_2, \lambda_3$. The $\lambda$'s were varied between $[0.4 - 1.5]$ in steps of $0.05$ and the performance on a separate test set of $10,000$ images were monitored. We start with the base cost function of Eqn. 1, and first select the optimum value of $\lambda_1$ by keeping $\lambda_2 = \lambda_3 = 1$. $\lambda_1$ value is fixed and we then determine optimal $\lambda_2$, and subsequently $\lambda_3$. Similarly for values of $\lambda_4, \lambda_5$, we start with the cost function of Eqn. 9, fix $\lambda_5 = 1$ and search for the optimum value of $\lambda_4$. Then we fix $\lambda_4$ and search for the optimal value of $\lambda_5$. 'Finally we search for the optimal value of $\lambda_6$ in Eqn. 12. The plots for the loss function with different values of $\lambda$ are shown in Figure 4.

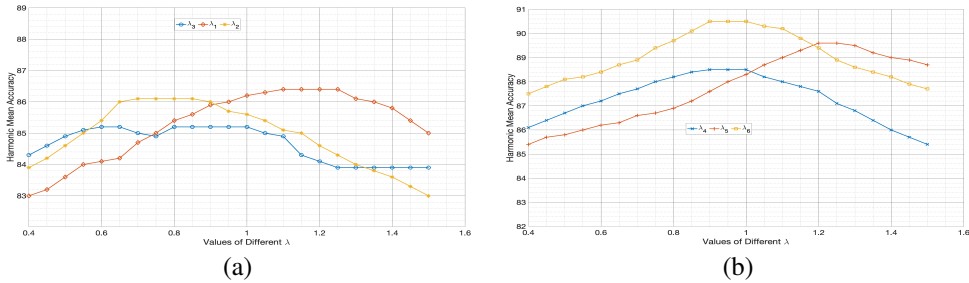

Figure 4: Hyperparameter Plots showing the value of $H$ and classification accuracy for different values of $\lambda$. The observed trends justify our final choice of the values.

## 5 CONCLUSION

We propose a multi-label GZSL approach for medical images. Our novel method can accurately synthesize feature vectors of unseen classes by learning a multi-label dictionary using graph aggregation and class specific features, alongwith text embedding relationships. Experimental results show our method outperforms other GZSL approaches in literature, and is consistently better across multiple public CXR datasets. Our approach is useful in scenarios where the number of disease classes are known but labeled samples of all classes cannot be accessed due to infrequent occurrence of such cases or lack of expert clinicians to annotate complex cases. While fully supervised settings still provide the best performance they are dependent upon sufficient labeled samples. Hence GZSL can be useful in addressing low data scenarios.

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
