# Multi-Label Generalized Zero Shot Chest Xray Classification Using Feature Disentanglement and Multi-Modal Dictionaries

## 1 Supplementary Material

### 1.1 Multi-Label GZSL For Natural Images

Following the two works on multi-label GZSL with natural images Lee et al. (2018); Huynh & Elhamifar (2020) we show results on the NUS-WIDE Chua et al. (2009), and Open Images Kuznetsova et al. (2020) datasets. The results are summarized in Table 1 and the reported metrics are precision (P), recall (R), and F1 score. Our method does better than these methods which demonstrates the advantages of using our feature disentanglement and graph aggregation strategies.

| GZSL and ablation results For natural Images | | | | | | |
|---|---|---|---|---|---|---|
| | NUS-WIDE | | | Open Images | | |
| Method | P | R | F1 | P | R | F1 |
| Lee et al. (2018) | 22.8 | 25.9 | 24.2 | 20.2 | 23.4 | 21.7 |
| Huynh & Elhamifar (2020) | 19.8 | 14.6 | 16.8 | 16.2 | 18.9 | 17.4 |
| **ML-GZSL** | **28.2** | **32.4** | **30.2** | **24.3** | **27.2** | **25.7** |
| $w\mathcal{L}_{agn-spec}$ | 26.2 | 30.8 | 28.4 | 23.9 | 26.2 | 25.0 |
| $w\mathcal{L}_{agn}$ | 27.6 | 31.1 | 29.3 | 24.3 | 26.6 | 25.4 |
| $w\mathcal{L}_{spec}$ | 25.8 | 29.4 | 27.5 | 22.0 | 25.1 | 23.5 |
| $w\mathcal{L}_{Dict_{Text}}$ | 26.0 | 28.9 | 27.4 | 22.5 | 26.0 | 24.2 |
| $w\mathcal{L}_{Dict_{Spe}}$ | 25.7 | 28.7 | 27.2 | 22.1 | 25.4 | 23.7 |
| $w\mathcal{L}_{ML-Syn}$ | 24.4 | 27.6 | 25.9 | 21.3 | 24.3 | 22.8 |

Table 1: **GZSL and ablation results For natural Images in Multi-Label setting:** Precision, Recall and F1-score are reported.

| | $\lambda_1$ | $\lambda_2$ | $\lambda_3$ | $\lambda_4$ | $\lambda_5$ | $\lambda_6$ |
|---|---|---|---|---|---|---|
| NIH Xray | 1.1 | 0.7 | 0.9 | 1 | 1.1 | 1.1 |
| CheXpert | 1.2 | 0.8 | 1.1 | 0.9 | 1.2 | 0.9 |
| NUS WIDE | 1.2 | 1 | 0.9 | 0.9 | 1.2 | 0.9 |
| Open Images | 1.3 | 1.1 | 0.9 | 1.2 | 1 | 1.1 |

Table 2: Values of all $\lambda$ for different datasets.

### 1.2 Realism

In order to evaluate the realism of synthetic features we let trained radiologists analyze images corresponding to the features. We reconstruct the xray images from the synthetic feature vectors using the feature disentanglement autoencoders' decoder part. We select 1000 synthetic images from 14 classes of the NIH dataset and ask two trained radiologists, having 12 and 14 years experience in examining chest xray images for abnormalities, to identify whether the images are realistic or not. Each radiologist was blinded to the other's answers.

Results for ML-GZSL show one radiologist ($RAD$ 1) identified 912/1000 (91.2%) images as realistic while $RAD$ 2 identified 919 (91.9%) generated images as realistic. Both of them had a high agreement with 890 common images (89.0% -"$Both\ Experts$" in Table 3) identified as realistic. Considering both $RAD$ 1 and $RAD$ 2 feedback, a total of 941 (94.1%) unique images were identified as realistic ("$Atleast\ 1\ Expert$" in Table 3). Subsequently, 59/1000 (5.9%) of the images were not identified as realistic by any of the experts ("$No\ Expert$" in Table3). Agreement statistics for other methods are summarized in Table 3.

| Agreement Statistics | Both Experts | Atleast 1 Expert | No Expert |
|---|---|---|---|
| ML-GZSL | **89.0** (890) | **94.1** (941) | **5.9** (59) |
| Lee Lee et al. (2018) | **85.1** (851) | **88.0** (880) | **12.0** (120) |
| Huynh Huynh & Elhamifar (2020) | **83.4** (834) | **85.1** (851) | **14.9** (149) |
| SDGN Wu et al. (2020) | **81.9** (819) | **83.9** (839) | **16.1** (161) |

Table 3: Agreement statistics for different image generation methods amongst 2 radiologists. Numbers in bold indicate agreement percentage while numbers within brackets indicate actual numbers out of 1000 samples.

```
[CLS], P, ##ne, ##um, ##oth, ##orax, [SEP]
```

Figure 1: Wordpiece tokenisation for *pneumothorax*. [CLS] is the classifier token and [SEP] denotes the end of the sentence.

## 1.3 EMBEDDINGS

We generate embeddings of image class labels using BioBERT Lee et al. (2020), a BERT Devlin et al. (2018)-like pre-trained model. BioBERT is pre-trained on biomedical literature, more specifically the model available from Huggingface[1], which is a base and cased model. To distill the label embeddings, each label textual representation is processed by BioBERT as explained below.

Each image text label is processed by wordPiece tokenisation Wu et al. (2016), which splits the text into tokens using punctuation and white spaces and then splits the tokens into wordpieces as shown in figure 1.

BERT models are composed of a stacked set of encoding multi-head self-attention layers Vaswani et al. (2017) (12 in our BioBERT model), which allows processing the left and right (bi-directional) context of a given token. The output of BERT is an embedding for each one of the input tokens, typically of 768 dimensions, and the total number of tokens identified by the tokeniser by the specific input. The output of this stack is a matrix of size $(Number\_of\_tokens \times Embedding\_size)$, where 768 is the typical size of the embeddings. Standard BERT architectures support processing text up to 512 tokens.

For our purposes, we consider a pooled set that produces a single vector with 768 dimensions. Pooling is performed by processing the output for the first classification [CLS] token by a fully connected layer (Linear) followed by a $tanh$ activation function as shown in equation 1.

$$pooling_{BioBERT} = tanh(Linear(biobert[CLS])) \tag{1}$$

Pre-training uses a Cloze task Taylor (1953), which trains BERT using a masked language. The embeddings generated from the image labels using BioBERT will be in a space in which similar concepts will be closer, e.g. using Cosine similarity. We envision using the similarities derived from these language models to guide or constrain the image feature generation algorithms.

The pooled embeddings still have 768 dimensions. Hayat et al. (2021) used a multi-layer perceptron and further adapted the embeddings to a lower dimension in a supervised fashion. In our work, do not fine-tune the language model and propose reducing the dimensionality by projecting the embeddings to a lower dimensional space using t-sne (t distributed stochastic neighbor embedding) Van der Maaten & Hinton (2008).

In t-sne, the distance between points in the original space (x) generated by the pooled BioBERT output is represented by a Gaussian distribution.

$$p_{ij} = \frac{exp(-||x_i - x_j||^2/2\sigma^2)}{\sum_{k \neq l} exp(-||x_i - x_l||^2/2\sigma^2)} \tag{2}$$

---

[1] https://huggingface.co/dmis-lab/biobert-v1.1

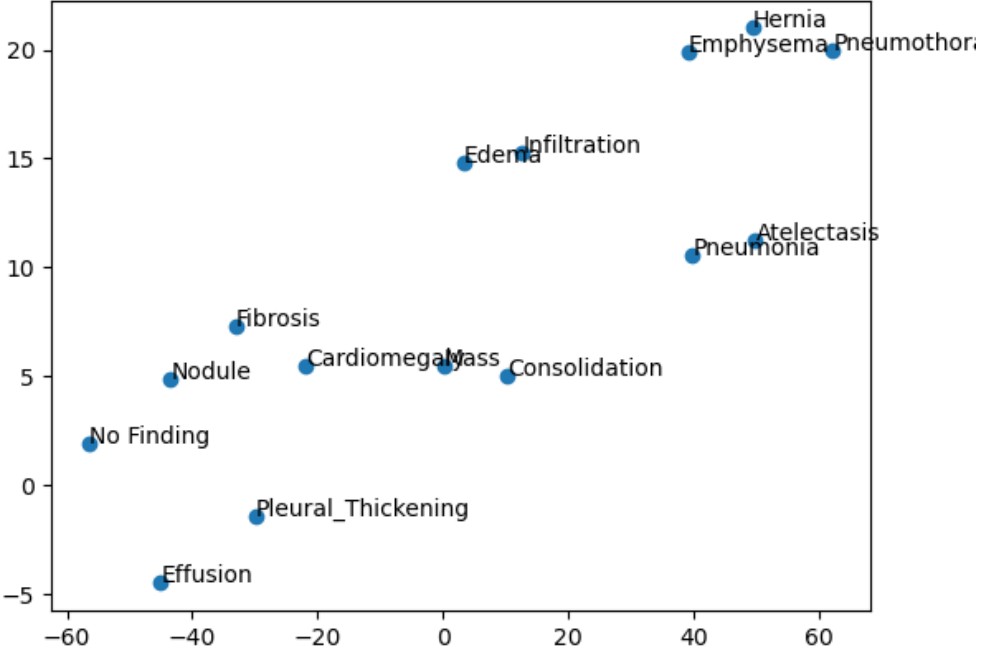

Figure 2: t-sne projection of BioBERT embeddings to 2 dimensions

The lower dimensional space (y) represents the distance using a t-student distribution.

$$q_{ij} = \frac{(1 + ||y_i - y_j||^2)^{-1}}{\sum_{k \neq l}(1 + ||y_k - y_l||^2)^{-1}} \quad (3)$$

Training consists in reducing the Kullback–Leibler divergence between the distribution of $p$ and $q$. As hyperparameters, we have set the dimensionality of the projected space (576, 384 and 192) and the perplexity (= 5).

Figure 2 shows a projection into a two dimensional space using the t-sne values. Table 1 of the main manuscript shows the actual cosine similarity values between all the 15 classes - 14 diseased classes and 'No Finding'. The $15 \times 15$ matrix in Table 1 of main manuscript has all diagonal elements equal to 1, which is obvious because it is the cosine similarity of a class's embedding with itself. During the feature generation stage we enforce constraints that the unseen class feature vectors should have cosine similarity values (with respect to other seen and unseen classes) close to the values shown in Table 1 of main manuscript. This matrix, which we refer as $Dict_{Text}$ - dictionary for text embeddings, is a realistic substitute for class attribute vectors. $Dict_{Text}$ avoids the labour intensive process of defining class attribute vectors and provides a quantitative relationship between different disease labels.

## 1.4 VISUALIZATION OF SYNTHETIC IMAGE FEATURES

Figure 3 (a) shows a t-SNE plot of features of actual data from the NIH chest X-ray dataset where the different classes are spread over a wide area, with slight overlap between some classes. Figure 3 (b) shows the distribution of synthetic features generated by our method. Although the corresponding clusters for the different classes have separate locations in the two figures they are similar to that of Figure 3 (a) in the sense that the different classes are similarly separated and there is minimal overlap.

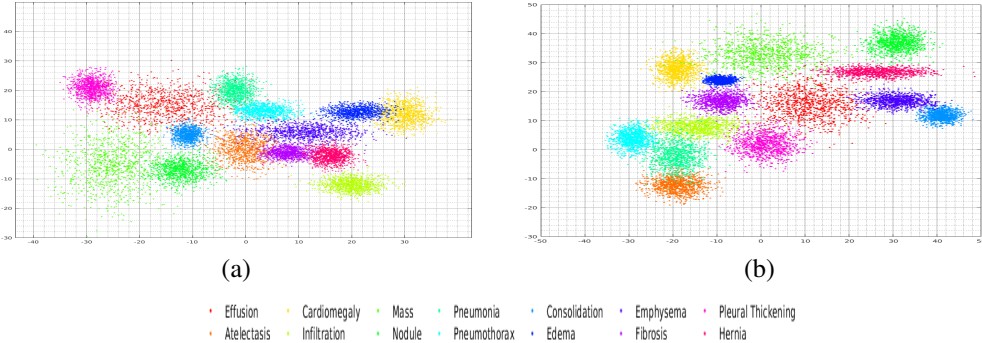

Figure 3: Feature visualizations for NIH Chest X-ray Dataset: (a) All classes from actual dataset; distribution of synthetic samples generated by (b) ML-GZSL. Different colours represent different classes. (b) is similar to (a) in terms of the clusters being separate with minimal overlap.