# OpenReview forum: "Multi-Label Generalized Zero Shot Chest Xray Classification  Using Feature Disentanglement and Multi-Modal Dictionaries"
_ICLR.cc/2024/Conference — ICLR 2024 Conference Withdrawn Submission_

### Official Review · Reviewer_hByk · 2023-10-26

**Soundness:** 1 poor
**Presentation:** 1 poor
**Contribution:** 1 poor
**Rating:** 1
**Confidence:** 5

**Summary:**

The paper presents a multi-label zero-shot Chest X-ray classification method, with many ideas being explored in a recent work [1].



[1] Mahapatra, D., Jimeno Yepes, A. J., Kuanar, S., Roy, S., Bozorgtabar, B., Reyes, M., & Ge, Z. (2023, October). Class Specific Feature Disentanglement and Text Embeddings for Multi-label Generalized Zero Shot CXR Classification. In International Conference on Medical Image Computing and Computer-Assisted Intervention (pp. 276-286). Cham: Springer Nature Switzerland.

**Strengths:**

Many ideas have been validated by a published work.

**Weaknesses:**

Many ideas have been validated by a published work.

**Questions:**

None

---

### Official Review · Reviewer_9scs · 2023-10-29

**Soundness:** 1 poor
**Presentation:** 1 poor
**Contribution:** 1 poor
**Rating:** 1
**Confidence:** 4

**Summary:**

This paper focuses on multi-label generalized zero shot learning on chest xray datasets. The propose method is the same as MICCAI'23 paper "Class Specific Feature Disentanglement and Text Embeddings for Multi-label Generalized Zero Shot CXR Classification" with modification on feature generation network from Mixup to WGAN. Experiments are performed under chest xray datasets: Chest X-ray14 and CheXpert.

**Strengths:**

No

**Weaknesses:**

This paper content is largely overlapped with MICCAI'23 paper. The similarities are too many, including introduction, contribution points, methods and even experiment results. What is even more controversial is that, with a complex WGAN introduced in this paper, replacing Mixup in MICCAI'23 version, yet Table 2 experiment results are basically the same.

I have a serious doubt for this paper reproducibility and originality. I vote for strong reject.

**Questions:**

.

**Details Of Ethics Concerns:**

Highly similar paper MICCAI'23 "Class Specific Feature Disentanglement and Text Embeddings for Multi-label Generalized Zero Shot CXR Classification". No reference found.

---

### Official Review · Reviewer_Lpe4 · 2023-10-31

**Soundness:** 2 fair
**Presentation:** 2 fair
**Contribution:** 1 poor
**Rating:** 3
**Confidence:** 4

**Summary:**

The authors suggest a multi-modal, multi-label generalized zero-shot learning (GZSL) method for medical images. The method synthesizes features of unseen classes by utilizing multi-modal dictionaries and feature disentanglement. The proposed method outperforms state-of-the-art GZSL algorithms for chest X-rays.

**Strengths:**

- Attempt to tackle the challenging task of synthesizing features for unseen classes.
- The motivation is clearly stated and the paper is easy to follow.

**Weaknesses:**

- The technical contribution of the paper is critically limited.
- The experiments are limited to datasets from chest X-rays. To back up the claim that the proposed model is robust, experiments could be expanded to include more datasets, such as skin lesion images.
- The manuscript has a problem with plagiarism (please see the Ethics Concerns for more details).
- Minor but a few typos, such as 'X-ray' (it is not 'x-ray').

**Questions:**

Could the authors precisely explain how their work is different from the following paper?

[1] Mahapatra, D. et al. (2023). Class Specific Feature Disentanglement and Text Embeddings for Multi-label Generalized Zero Shot CXR Classification. In: Greenspan, H., et al. Medical Image Computing and Computer Assisted Intervention – MICCAI 2023. MICCAI 2023. Lecture Notes in Computer Science, vol 14221. Springer, Cham.

**Details Of Ethics Concerns:**

A substantial portion of the content in this paper is repeated or slightly paraphrased from a previously published work [1]. I could not even find any references for the paper. Because of this, I think the work has a plagiarism or duplicate submission problem.

[1] Mahapatra, D. et al. (2023). Class Specific Feature Disentanglement and Text Embeddings for Multi-label Generalized Zero Shot CXR Classification. In: Greenspan, H., et al. Medical Image Computing and Computer Assisted Intervention – MICCAI 2023. MICCAI 2023. Lecture Notes in Computer Science, vol 14221. Springer, Cham.